 **LLM Reasoners**

# New Evaluation, Library, and Analysis of Step-by-Step Reasoning with Large Language Models

**Shibo Hao**[1*], **Yi Gu**[1*], **Haotian Luo**[1*], **Tianyang Liu**[1],
**Xiyan Shao**[1], **Xinyuan Wang**[1], **Shuhua Xie**[1], **Haodi Ma**[2],
**Adithya Samavedhi**[1], **Qiyue Gao**[1], **Zhen Wang**[1,3], **Zhiting Hu**[1]
[1]UC San Diego, [2]University of Florida, [3]MBZUAI
{s5hao, yig025, zhh019}@ucsd.edu
https://www.llm-reasoners.net/

## Abstract

Generating accurate step-by-step reasoning is essential for Large Language Models (LLMs) to address complex problems and enhance robustness and interpretability. Despite the flux of research on developing advanced reasoning approaches, systematically analyzing the diverse LLMs and reasoning strategies in generating reasoning chains remains a significant challenge. The difficulties stem from the lack of two key elements: **(1)** an automatic method for evaluating the generated reasoning chains on different tasks, and **(2)** a unified formalism and implementation of the diverse reasoning approaches for systematic comparison. This paper aims to close the gap: **(1)** We introduce AutoRace for fully automated reasoning chain evaluation. Existing metrics rely on expensive human annotations or pre-defined LLM prompts not adaptable to different tasks. In contrast, AutoRace automatically creates detailed evaluation criteria tailored for each task, and uses GPT-4 for accurate evaluation following the criteria. **(2)** We develop LLM Reasoners, a library for standardized modular implementation of existing and new reasoning algorithms, under a unified formulation of the *search*, *reward* and *world model* components. With the new evaluation and library, **(3)** we conduct extensive study of different reasoning approaches (e.g., CoT, ToT, RAP). The analysis reveals interesting findings about different factors contributing to reasoning, including the reward-guidance, breadth-vs-depth in search, world model, and prompt formats, etc.

## 1 Introduction

A central topic in Large Language Model (LLM) research is to enhance their ability of complex reasoning on diverse problems (e.g., logical reasoning, mathematical derivations, and embodied planning). Rich research has been done to generate multi-step reasoning chains with LLMs, such as Chain-of-Thoughts (CoT, Wei et al., 2022), Tree-of-Thoughts (ToT, Yao et al., 2023), Reasoning-via-Planning (RAP, Hao et al., 2023a), among others (Zhu et al., 2022; Xie et al., 2023; Zhuang et al., 2023; Khalifa et al., 2023; Creswell & Shanahan, 2022). However, despite the burgeoning body of literature, there lacks a systematic analysis and understanding of the diverse approaches, mainly due to two key challenges:

**First, *automatic evaluation* of multi-step reasoning chains is difficult.** Previous studies mostly rely on the accuracy of the final answers as a proxy for assessing the reasoning processes. However, as LLMs tend to produce unfaithful outputs or hallucinate, a correct final answer does not necessarily imply a logically sound reasoning chain (Figure 1, a) (Golovneva et al., 2022; Prasad et al., 2023; Tyen et al., 2023; Lyu et al., 2023; Liu et al., 2023). Indeed, by manually evaluating 100 reasoning chains generated by Llama-2-70B on the

---
[*]Equal contribution.

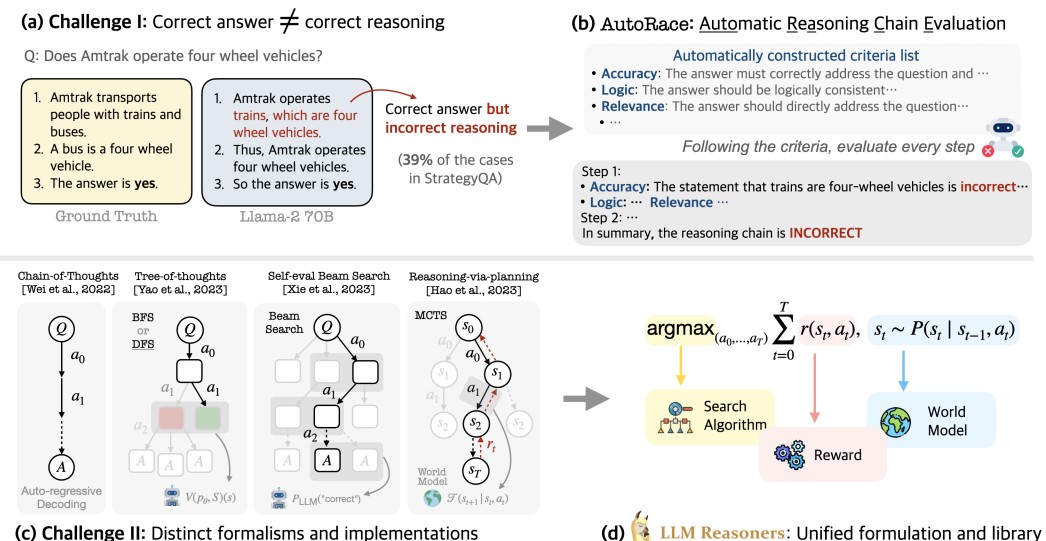

Figure 1: **(a)** The first challenge for analyzing step-by-step reasoning with LLMs: correct final answer may be derived from incorrect reasoning chains (*false-positive* chains), making it necessary to evaluate the reasoning chains directly. **(b)** Our proposed AutoRace for fully automated evaluation. **(c)** The second challenge stems from the diverse reasoning algorithms with seemingly distinct designs. **(d)** Our LLM Reasoners provides a unified formulation and standardized implementation.

StrategyQA questions (Geva et al., 2021), we found up to 39% of such *false-positive* cases that contain reasoning errors despite having correct final answers. Recent efforts have attempted to evaluate the reasoning chains directly, but often require non-trivial human efforts, such as human-written reasoning chains as references (Celikyilmaz et al., 2020), or manually-annotated datasets for training evaluation models (Golovneva et al., 2022; Prasad et al., 2023; Xia et al., 2024). He et al. (2023); Tyen et al. (2023) use GPT-4 to alleviate human cost, but still require demonstration questions and in-depth error analyses by human experts before applying to each new task. In addition, their instructions that prompt GPT-4 for evaluation are not adaptive to different tasks, leading to suboptimal performance (Section 3.2).

**Second, the varied reasoning approaches present distinct formalisms and implementations** (Figure 1, c). The disparity makes it difficult to analyze the nuanced differences of their reasoning chain generation and compare their critical design elements. Therefore, it is desirable to have a more holistic formulation and unified implementation. This would reveal the underlying connections among different approaches, and facilitate a more systematic comparison when combined with automatic reasoning evaluation discussed above.

To tackle the challenges, this paper proposes an automatic method for reasoning chain evaluation, develops a cohesive library for various latest reasoning approaches, and on this basis, performs extensive analysis of LLM step-by-step reasoning. More specifically, we first present **AutoRace (Automatic Reasoning Chain Evaluation)**, a fully automated approach for evaluating reasoning chains that adapts to different tasks without human efforts (Figure 1, b). For each reasoning task (e.g., math reasoning), AutoRace autonomously constructs a detailed *evaluation criteria list* by summarizing errors in LLM-generated reasoning chains. The criteria list is then used to instruct GPT-4 to evaluate any given reasoning chains on the task. Compared to the predefined human-written prompts (Tyen et al., 2023; He et al., 2023), the AutoRace criteria lists are automatically customized for each task with GPT-4 to ensure accurate evaluation. On a wide range of tasks, AutoRace shows strong correlation with human evaluation, and manages to detect 70.4% of incorrect reasoning chains that cannot be captured by the conventional final-answer-based evaluation.

We then introduce a unified perspective of reasoning algorithms, formulating them as a *search* process towards maximizing accumulated rewards (Figure 1, d). A wide range of existing reasoning algorithms can be interpreted as specific choices of the components in the unified formulation, including a *reward function r* to decide preferences on different reasoning steps, *world model $\mathcal{T}$* to specify the state transition, and *search algorithm* (e.g., beam search, Monte-Carlo tree search) to explore the expansive reasoning space. Based

on the unified perspective, we further develop the `LLM Reasoners` library that provides standardized implementation of these components with configurable options, plus rich LLM APIs and intuitive visualizations. As a result, `LLM Reasoners` allows us to easily reproduce the existing reasoning algorithms, and also compose new algorithms and apply to new tasks with minimal efforts.

With the new evaluation method and library, we conduct extensive analysis of reasoning chain generation of diverse LLMs and reasoning algorithms. We collect 6 challenging reasoning tasks that cover different reasoning skills (logical deduction, math, and embodied planning). Using a standardized evaluation protocol, we compare various most popular reasoning algorithms (e.g., CoT, ToT, RAP). The results offer a number of new insights into reasoning algorithm design—for example: (1) Reasoning as reward-guided search helps not only improve final accuracy, but also effectively alleviate false-positive reasoning chains; (2) For efficient search in the reasoning space, the breadth of search is generally more important than the depth for most tasks; (3) incorporating a world model that explicitly infers reasoning *state* would effectively improve the LLM reasoning ability, particularly for tasks in embodied environments; (4) inappropriate prompt format design might inadvertently lead to false-positive reasoning chains. We also compare across diverse LLMs (GPT-4, Claude-3, Gemini, etc.) on their CoT reasoning chains. We release all code and experiments of `AutoRace` and `LLM Reasoners` at https://www.llm-reasoners.net/, hoping to spur the progress of research on LLM complex reasoning.

## 2 Related Work

**Evaluation of Reasoning Chains.** Traditionally, to evaluate the reasoning process, generated reasoning chains are compared with human-written explanations, which is known as reference-based reasoning evaluation. Conventional natural language generation (NLG) metrics were applied to calculate the similarity between machine-generated chains and human-crafted ones (Celikyilmaz et al., 2020; Clinciu et al., 2021; Welleck et al., 2022). Towards reference-free reasoning evaluation, Dalvi et al. (2021); Saparov & He (2022); Han et al. (2022) designed structured reasoning tasks so that the reasoning process can be checked by a program automatically. Recently, ROSCOE (Golovneva et al., 2022) and ReCEval (Prasad et al., 2023) proposed reference-free metrics on general domains, measuring similarity, informativeness and correctness among steps. With the rapid development of LLM, Tyen et al. (2023) proposed to prompt GPT-4 with few-shot demonstrations, and Liu et al. (2023) experimented with knowledge-enhanced prompting by providing LLMs with relevant information. However, the results indicate that it's still challenging for LLMs to evaluate reasoning chains. He et al. (2023) crafted a detailed instruction inspired by the Socratic method, but the method requires GPT-4 to generate a reference chain at first, limiting its performance for challenging reasoning tasks that GPT-4 fails to solve. Besides, the fixed prompt template is not adjustable to different tasks, which also leads to suboptimal evaluation accuracy. In this work, we focus on LLM-based reference-free reasoning chain evaluation. Our method is generally more accurate and robust than existing metrics, while also saving any additional human efforts. Concurrent to our work, Xia et al. (2024) focused on the false postivie problem in mathmatical reasoning with a fine-tuned LLM as the evaluator. Paul et al. (2024) attempted to measure and improve the faithfulness of reasoning with causal inference.

**Step-by-step Reasoning with LLMs.** A common practice to enhance the reasoning with LLMs is to generate intermediate reasoning steps, employing methods such as chain-of-thought prompting (Wei et al., 2022; Kojima et al., 2022) or question decomposition (Zhou et al., 2022; Li et al., 2023). Inspired by the deliberate reasoning of humans, recent research has focused on searching for better reasoning chains guided by reward (Zhu et al., 2022; Xie et al., 2023; Yao et al., 2023; Zhuang et al., 2023; Khalifa et al., 2023; Creswell & Shanahan, 2022). Hao et al. (2023a) proposed to incorporate a world model into reasoning, which simulates the state of the world. This enables LLMs to reason in a manner close to humans' conscious planning. Hu & Shu (2023) presented the LAW formulation that connects the concepts of language models, agent models, and world models for more advanced and robust reasoning. Xiang et al. (2024) delivered a multi-modal model that takes into natural language instruction and generate video next state, towards the goal of a general world

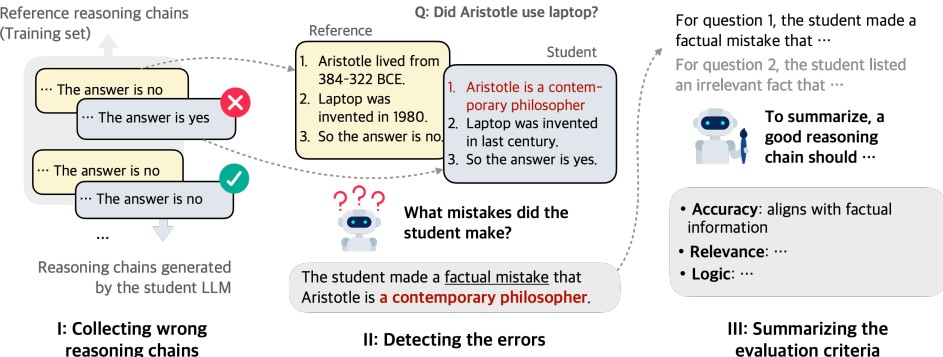

Figure 2: For any reasoning tasks (e.g., commonsense reasoning), AutoRace automatically constructs an evaluation criteria list to help itself evaluate reasoning chains in this domain.

model for machine reasoning. Another line of works fine-tune the LLM as a reasoning policy that maximize or sample proportional to reward (Shao et al., 2024; Yu et al., 2024). We include a more systematic summary of reasoning algorithms in Section 4. Related to the scope of this paper, recent works (Welleck et al., 2024; Chen et al., 2024) also surveyed and analyzed tree search algorithms for reasoning.

## 3  AutoRace: Automatic Reasoning Chain Evaluation

In this section, we present AutoRace that offers more insights into the LLM reasoning process than final answer correctness (Figure 1). Compared to previous works (Tyen et al., 2023; He et al., 2023) that prompt GPT-4 with fixed human-written instructions, AutoRace involves a "learning" process, which helps it to adapt to any problem domains. Specifically, for each task, AutoRace automatically collects LLM-generated incorrect reasoning chains, and summarizes evaluation criteria from them (Figure 2). With the criteria, GPT-4 can pay more attention to common errors for this certain domain, and make a more accurate evaluation. Compared to previous works that train an evaluation model (Golovneva et al., 2022; Prasad et al., 2023) by optimization model parameters, AutoRace effectively leverages GPT-4's strong prior knowledge, so that it is able to learn from only incorrect reasoning chains, which can be collected automatically.

### 3.1  Evaluation Method

To formulate the problem, we consider a reasoning question $x$, and LLM-generated reasoning chains $z$, and the predicted answer $y$. Additionally, we have the reference answer $y_r$, accompanied by a reference reasoning chain $z_r$, which are available in the training set $D_{train}$ of most existing reasoning datasets. Our goal is to develop an automatic evaluation metric for the reasoning chain, $s(z) \in \{0, 1\}$, which is better aligned with human evaluation of the reasoning chains.

As the first step to criteria list construction, one needs to find out what kinds of errors are common for a task. Therefore, AutoRace is designed to condense the criteria from real mistakes in LLM-generated reasoning chains (Figure 2, I). Here, we make use of the fact that a reasoning chain reaching a wrong answer must include an intermediate mistake. Given a sub-sampled training set $D = \{(x, y_r, z_r)\} \subset D_{train}$, we run Chain-of-Thoughts reasoning with an LLM (referred as the student LLM) to expand the dataset to $D' = \{x, y_r, z_r, y, z\}$, where $z$ is the reasoning chain generated by the student and $y$ is the predicted answer extracted from $z$. Then, we can filter out a subset where the generated answers disagree with the reference answers, $D_{\text{error}} = \{(x, y_r, z_r, y, z) \in D' \mid y_r \neq y\}$.

Having these reasoning chains with errors, the next goal is to compile a criteria list. To reduce the difficulty, we divide it into two simple steps: *Detection* and *Summarization*. The Detection step identifies the specific errors in a reasoning chain. GPT-4 is presented with the question, the reference reasoning chain, and the student reasoning chain. It is then instructed to point to the mistake in the student reasoning chain (Figure 2, II). The underlying rationale is that, even if the question $x$ might be challenging for GPT-4 to solve on its own, it has a good chance of understanding the question and identifying the mistakes once it has access to the reference reasoning chain.

| Method | Math | | Common | | Logical | | Average | Fully Auto. |
|---|---|---|---|---|---|---|---|---|
| | GSM8k | Arith | Strategy | Cosmos | Logic | Sort | | |
| *Answer-based* | 0.94 | 0.94 | 0.76 | 0.67 | 0.87 | 0.94 | 0.85 | |
| SocREval | 0.89 | 0.85 | 0.71 | 0.80 | 0.89 | 0.77 | 0.82 | ✗ |
| Direct (trace) | 0.90 | 0.38 | 0.80 | 0.72 | 0.21 | 0.36 | 0.56 | ✗ |
| Direct (step) | 0.85 | 0.43 | 0.83 | 0.73 | 0.75 | 0.33 | 0.65 | ✗ |
| CoT (step) | 0.78 | 0.74 | 0.78 | 0.72 | 1.00 | 0.86 | 0.81 | ✗ |
| AutoRace (Ours) | 0.91 | 0.85 | 0.79 | 0.78 | 0.97 | 0.86 | **0.86** | ✓ |

Table 1: Evaluation accuracy of various reasoning chain evaluation metrics. We also list the accuracy of answer-based evaluation as a reference. Note that AutoRace is the only metric that does not take any human inputs specific to reasoning tasks (i.e., fully automated). We highlight the best reasoning chain metrics (dark green) and metrics within 5% of the best performance (light green) for each task. AutoRace achieves the best average accuracy and is robust across datasets.

After collecting the errors in reasoning chains, GPT-4 is prompted to summarize these specific instances into a criteria list (Figure 2, III). Eventually, GPT-4 is able to evaluate any new reasoning chain $z$ given a question $x$, by checking each criteria on each reasoning step. The prompt template of each phrase is in Appendix A.7.

## 3.2 Experiments

To measure the efficacy of reasoning chain evaluation metrics, we use human-annotated binary labels of reasoning chains as the ground truth, and calculate their accuracy.

**Datasets.** We experiment on 6 datasets covering mathematical, commonsense and logical reasoning. 5 of them are from previous works (Golovneva et al., 2022; Tyen et al., 2023), originating from GSM8K (Cobbe et al., 2021), Multistep-Arithmetics (Srivastava et al., 2023), DROP (Dua et al., 2019), COSMOS-QA (Huang et al., 2019), Logical-Deduction (Srivastava et al., 2023) and Word-Sorting (Srivastava et al., 2023). We additionally sample and manually label reasoning chains from StrategyQA (Geva et al., 2021). The detailed statistics of these datasets can be found in Appendix A.

**Baselines.** We compare AutoRace with other LLM-based evaluation metrics for reasoning chains. (a) **SocREval** (He et al., 2023) crafted a detailed instruction prompt for GPT-4 through the Socratic method, which includes asking it to generate a reference reasoning chain before evaluation. This method also requires a one-shot demonstration written by humans for each task. Tyen et al. (2023) proposed three methods: (b) **Direct (trace)** asks GPT-4 to directly evaluate a reasoning chain; (c) **Direct (step)** asks GPT-4 to check the reasoning step by step; (d) **CoT (step)** asks GPT-4 to generate a reasoning process before evaluating each reasoning step. All these methods require 3-shot demonstrations written by humans. We don't experiment with metrics based on fine-tuned small models (Golovneva et al., 2022; Prasad et al., 2023), as existing literature has already indicated a substantial performance gap between these methods and LLM-based metrics (He et al., 2023).

**Results.** We collect 4 incorrect reasoning chains for each task to create the criterion list. The results are presented in Table 1. We can observe that among all metrics for reasoning chains, AutoRace achieves the best overall performance. It excels in 3 out of 6 tasks and exhibits robustness, maintaining performance levels within 5% of the best results across the board. Note that different from all baseline, which requires human-written demonstrations, AutoRace does not need any human input specific to reasoning tasks. Indicated by the confusion matrix (Figure 3, left), AutoRace is good at detecting incorrect reasoning chains, without sacrificing the performance in correct reasoning chains. On the contrary, SocREval mistakenly classifies many correct reasoning chains to be incorrect. Since SocREval asks GPT-4 to generate its own response as the reference, whenever GPT-4 fails to solve the problem itself, it's very likely to evaluate the reasoning chain to be incorrect, misled by the wrong reference. AutoRace enables GPT-4 to evaluate reasoning chains on problems that it fails to solve itself, as a case study shown in Figure 6. Specifically, in a problem from MultiArith, a task for testing multi-digit arithmetic, SocREval fails because GPT-4 generates the reference with the same mistakes as the reasoning chain to be evaluated, but AutoRace identifies the subtle errors with the detailed criteria. We include more detailed results in

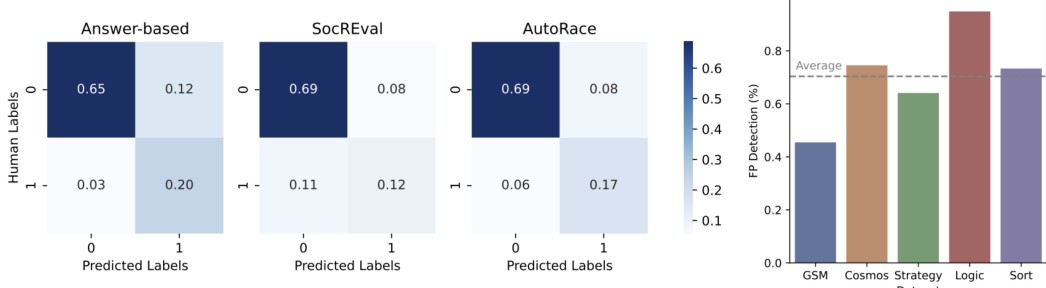

Figure 3: Analysis on different reasoning chain evaluation methods: (Left) The macro-averaged confusion matrix of these methods. SocREval and AutoRace are both good at detecting incorrect reasoning chains, while SocREval mistakenly classifies correct reasoning chains as wrong more frequently. (Right) AutoRace can recognize 70.4% of the false positive reasoning chains, showing the promise to be a great complement to answer accuracy.

Appendix A, that indicate AutoRace is robust to the number of incorrect reasoning chains used for construct criteria construction, the required cost for API call is reasonable, and the criteria lists are transferable across tasks to a certain extent.

Moreover, when compared with the answer-based metric, AutoRace also outperforms it in 3 of 6 tasks and exhibits better overall performance. We additionally calculate the accuracy of AutoRace on reasoning chains with mistakes but reaching a correct answer (false positive reasoning chains), and it turns out that AutoRace managed to detect 70.4% of the false positive reasoning chains averaged across different tasks (Figure 3, right). We examine some false positive reasoning chains detected by AutoRace, and find the explanation given by AutoRace is mostly reasonable. The false positive reasoning chains can be classified into 3 types (Table 10). Based on these results, we believe AutoRace would be a useful metric complementary to answer-based evaluation.

## 4 LLM Reasoners: A Unified Formulation and Library

Besides reasoning chain evaluation, another difficulty in the analysis of reasoning algorithms lies in their distinct formulations and implementations. To investigate the critical design elements that affect the nuanced performance, we aim to deliver a more holistic formulation (Section 4.1) and unified implementation (Section 4.2) in this section.

### 4.1 Unified Formulation

There has been rich research on constructing reasoning chains to solve problems using LLMs, from the simplest CoT prompting (Wei et al., 2022), to tree search algorithms guided by a reward function (Yao et al., 2022; Xie et al., 2023; Hao et al., 2023a) and a world model (Hao et al., 2023a). These methods, among many others, can be formulated as a search process that maximizes the accumulated **reward** $\arg\max_{(a_0,...,a_T)} \sum_{t=0}^{T} r(s_t, a_t)$, with a **world model** that predicts state transition $s_t \sim \mathcal{T}(\cdot|s_{t-1}, a_{t-1})$, and a **search algorithm** to optimize the objective. This section elaborates on these three crucial components and demonstrates how recent reasoning algorithms can be interpreted as special cases within this framework, with specific choices of these three components (Table 2).

**World model.** The world model defines the state transition distribution $\mathcal{T}(s_t|s_{t-1}, a_{t-1})$. For example, we can formulate the reasoning state of CoT (Wei et al., 2022) as the list of all previous actions, i.e., $s_t = (a_0, a_1, ..., a_{t-1})$, and thus the world model represents a deterministic transition which always appends an action to the action history. Beyond this trivial definition, recent studies seek a more substantive depiction of the reasoning state, e.g., the description of the physical environment, or the set of known variables, etc. To track the reasoning state, Liu et al. (2022); Guan et al. (2023) augment LLMs with a physical engine or PDDL domain model. Li et al. (2023) train a model to predict the entity states as a latent variable, and RAP (Hao et al., 2023a) apply the LLM as general world model for reasoning. When the LLM can interact with the external environment, e.g., calling tools (Zhuang et al.,

| Method | Reward $r$ | World Model | Search Alg. |
|---|---|---|---|
| CoT | $p_{LLM}(a_t|a_0, ..., a_{t-1}, P_{CoT})$ | $s_t = (a_0, ..., a_{t-1})$ | Gready |
| ToT | $p_{LLM}("correct" \mid a_0, ..., a_t, P_{self\_eval})$ | $s_t = (a_0, ..., a_{t-1})$ | BFS/DFS |
| Self-Eval | $p_{LLM}("correct" \mid a_0, ..., a_t, P_{self\_eval})$ | $s_t = (a_0, ..., a_{t-1})$ | Beam search |
| Toolchain* | LST, self-consistency, etc. | $s_t = f_{tool}(s_{t-1}, a_{t-1})$ | A* search |
| ORM | $f_{ORM}(s_t, a_t)$ if $t = T$, else 0 | $s_t = (a_0, ..., a_{t-1})$ | Rand. Shoot. |
| PRM | $f_{PRM}(s_t, a_t)$ | $s_t = (a_0, ..., a_{t-1})$ | Rand. Shoot. |
| SitSup | $f_{finetuned}(s_t, a_t)$ | $s_t \sim p_{finetuned}(\cdot \mid s_{t-1}, a_{t-1})$ | Greedy |
| RAP | Likelihood, self-eval., etc. | $s_t \sim p_{LLM}(\cdot \mid s_{t-1}, a_{t-1})$ | MCTS |

Table 2: Representative reasoning algorithms, including CoT (Wei et al., 2022), ToT (Yao et al., 2023), Self-Eval (Xie et al., 2023), Toolchain* (Zhuang et al., 2023), ORM (Cobbe et al., 2021), PRM (Lightman et al., 2023), SitSup (Li et al., 2023), RAP (Hao et al., 2023a), summarized in terms of the reward function, world model, and search algorithm.

2023), the environment is the world model. For prompt optimization (Wang et al., 2023b) or adversarial attack (Guo et al., 2024), the target LLM also performs the role of a world model as it provides a feedback to a prompt. Recent work started to develop multi-modal world models at scale, such as GAIA-1 (Hu et al., 2023) for auto-driving, UniSim (Yang et al., 2023) for robotic manipulation, Genie (Bruce et al., 2024) for 2D games, and Pandora (Xiang et al., 2024), towards a general world model that generates next video state given natural language as inputs.

**Reward function.** The reward function $r(s_t, a_t)$ decides whether a reasoning step is desired. CoT implicitly employs the likelihood predicted by the language models as the reward, as it generates the next reasoning step with high likelihood conditioned on previous steps and a CoT prompt, i.e., $r(s_t, a_t) = p_{LLM}(a_t|a_0, ..., a_{t-1}, P_{CoT})$. Yao et al. (2023); Hao et al. (2023a); Xie et al. (2023); Ouyang et al. (2023) propose to use self-evaluation as the reward, asking the LLM to choose if the last action is "correct" or "wrong", or output a confidence score with a self-evaluation prompt, e.g., $r(s_t, a_t) = p_{LLM}("correct" \mid s_t, a_t, P_{self-eval})$. Cobbe et al. (2021); Paul et al. (2023); Yuan et al. (2024) train outcome-supervised reward models (ORMs) to evaluate a reasoning chain, which predicts the reward for a complete reasoning chain. More recent works (Khalifa et al., 2023; Lightman et al., 2023; Sun et al., 2024; Wang et al., 2023a) train step-by-step reward models with process supervision (PRM), to provide a more accurate reward for every step. One can also define task-specific heuristic functions as rewards (Zhuang et al., 2023; Hao et al., 2023a).

**Search Algorithm.** The expansive reasoning space makes exhaustive search infeasible and calls for the use of more efficient search algorithms. For example, CoT implicitly applies greedy decoding for the reasoning step with the highest reward[1]. Another common technique is to sample multiple reasoning chains, and return the one with the highest accumulated reward (Cobbe et al., 2021; Lightman et al., 2023; Wang et al., 2023a), which in essence is a random shooting algorithm (Kothare et al., 1996). Other widely used search algorithms include DFS (Yao et al., 2023), beam search (Xie et al., 2023), A* (Zhuang et al., 2023), and MCTS (Hao et al., 2023a; Zhao et al., 2024; Chi et al., 2024). An alternative paradigm is learning a policy model to maximize the reward (Havrilla et al., 2024; Shao et al., 2024) with RL, or sample proportional to reward (Yu et al., 2024) for diverse reasoning.

### 4.2 Library Design

`LLM Reasoners` implements our unified formulation for multi-step reasoning with a modular design. As illustrated in Figure 7, users can easily set up a reasoning method by defining the `WorldModel` and `SearchConfig`, and importing a `SearchAlgorithm`. Building on these three main base classes, `LLM Reasoners` has included new components to augment reasoning, e.g., pre-trained reward models (Yuan et al., 2024), tool-calling modules (Yao et al., 2022; Hao et al., 2023b), and new examples like scientific reasoning, e.g., for chemistry (Ouyang et al., 2023). We have also integrated rich LLM APIs, standard evaluation pipelines, and a general interactive visualization tool `visualizer`. More details are shown in Appendix B.

---

[1]It's usually implemented as token-level greedy decoding.

| Method | Math | | | Logical | Common | Embodied |
|---|---|---|---|---|---|---|
| | GSM8k* | AQuA* | Game24 | PrOnto | StrategyQA* | Blocks |
| CoT | 0.37 (0.54) | 0.09 (0.34) | 0.04 | 0.58 | 0.34 (0.76) | 0.05 |
| ToT (BFS) | 0.53 (0.58) | 0.15 (0.42) | 0.04 | 0.52 | 0.41 (0.76) | 0.09 |
| ToT (DFS) | 0.45 (0.52) | 0.10 (0.36) | **0.07** | 0.44 | **0.42** (0.76) | 0.08 |
| RAP | **0.58 (0.64)** | **0.20 (0.47)** | **0.07** | **0.59** | 0.28 (**0.77**) | **0.51** |

Table 3: Experimental results of various reasoning methods on every dataset. On three datasets marked with ∗, we evaluate with `AutoRace`, and also show the answer-based (in brackets) for reference. On other datasets, we evaluate the reasoning chain with oracle verifiers (e.g., a rule-based program, or a simulator) due to their nature of close domains. The best method in every metric is highlighted in **bold**.

## 5 Analysis of LLM Step-by-step Reasoning

To better understand multi-step reasoning algorithms and analyze the design elements critical to better reasoning performance, we evaluate them on diverse reasoning datasets, utilizing our `AutoRace` metric and `LLM Reasoners` library.

### 5.1 Datasets

For a comprehensive evaluation, we first collect reasoning tasks of several categories, where each category requires different reasoning skills.

**Mathematical Reasoning.** We select (1) GSM8k (Cobbe et al., 2021), a popular dataset of math word problems that requires understanding the relationship between numbers and multiple steps of mathematical calculation; (2) AQuA (Ling et al., 2017), which additionally requires the skill to perform algebra operations. Both answer-based and `AutoRace` metrics are employed on these two datasets. We also include the famous (3) *Game of 24*, following the settings in Yao et al. (2023). This task requires constructing an equation with four given numbers and basic arithmetic operations. We evaluate the reasoning chain on *Game of 24* with a program.

**Commonsense Reasoning.** We take StrategyQA (Geva et al., 2021) as the commonsense reasoning dataset. Each sample is an open-domain yes-no question that requires raising related commonsense knowledge and multiple steps of inference. We evaluate the reasoning chain using both answer-based and `AutoRace` metrics.

**Logical Reasoning.** The tasks involve a set of logical principles, initial statements, and a concluding hypothesis. The challenge is to perform multi-step deductions to determine if the final hypothesis is true. We use PrOntoQA (Saparov & He, 2022) in this category. The evaluation is based on a rule-based program, since the problems are from a close domain.

**Embodied Planning.** The capability of LLMs to power embodied agents presents an interesting area of study, as it involves the understanding of the physical world, and requires a strong planning ability toward the goal. To assess this capacity for embodied planning, we employ the Blocksworld benchmark (Valmeekam et al., 2023), where an agent must reach a specific block stacking arrangement through moving operations such as `PickUp` and `Stack`. For evaluation, we examine whether the generated chain is valid and can lead to the target state with a simulator.

### 5.2 Evaluating Reasoning Algorithms

**Compared methods.** To analyze the connections between recent step-by-step reasoning methods, we pick three representative methods, CoT (Wei et al., 2022), ToT (Yao et al., 2023), and RAP (Hao et al., 2023a). Different CoT which autoregressively decodes the reasoning chain, ToT and RAP define the reward and include tree search algorithms. RAP additionally incorporates an explicit world model.

**Configurations.** For ToT and RAP, we mainly apply the combination of two rewards: (1) Self-evaluation: Prompting the LLMs to evaluate the new action, and use the logits of "good" as the reward, $P_\theta(\texttt{"Good"} \mid s, a)$. (2) Likelihood: Calculating the log-likelihood of predicting the next action given the current state, i.e., $P_\theta(a \mid s)$. The definition of states and the world

model for RAP depends on the reasoning tasks. Benefiting from the explicit world model, there are also other rewards available for RAP on certain tasks. More details about RAP implementation are described in Appendix C.1.

**Implementation details.** To ensure reproducibility and accessibility, we use one of the leading open-sourced LLMs, Llama 2 70B (Touvron et al., 2023b), quantized with GPT-Q (Frantar et al., 2022) for all tasks and methods. To make a fair comparison, we restrict search-based methods to explore up to 10 reasoning chains: The breadth limit is 10 for ToT (BFS), the maximum number of visited terminal nodes is 10 for ToT (DFS), and the maximum number of iterations is 10 in RAP, which is based on Monte-Carlo Tree Search. We manually crafted 10 examples of reasoning chains for each task, and all methods share this same example pool. For each test case, 4 examples are randomly sampled to form the demonstrations in the prompt, resulting in a 4-shot learning setting.

### 5.2.1 Results

Table 1 shows a comparative analysis of step-by-step reasoning algorithms. Overall, ToT consistently outperforms the vanilla CoT, and RAP further improves upon ToT. Notably, the evaluation metric (`AutoRace`) is generally lower than the answer-based metric, especially in the AQuA and StrategyQA datasets. This discrepancy suggests significant potential for enhancements in future reasoning algorithms. We outline several key findings below.

**Reward-guided Search Reduces False Positives.** Enhanced exploration in the reasoning space, facilitated by effective reward functions, naturally leads to the superior performance of search-based methods (ToT and RAP) over the autoregressive decoding method CoT. However, a noteworthy observation is that these search-based methods also yield fewer false positive reasoning chains, indicated by the smaller gap between `AutoRace` and answer-based metric (Table 3), and higher `AutoRace` score of reasoning chains with correct answers (Figure 4). Further examination of examples with ToT (BFS) reveals that it effectively avoids some false positives by discarding reasoning steps with low rewards. In contrast, CoT lacks this mechanism to "regret". E.g., in type-A false positive chains made by CoT (Table 10), while some reasoning mistakes are identifiable by the LLM itself, CoT fails to amend errors from previous steps, only able to overlook them in the following steps.

**Importance of Search Breadth Over Depth.** By comparing two variants of ToT, ToT (BFS) and ToT (DFS), our results show that BFS is relatively better on two math word problems, logical and embodied tasks. This indicates that when the search space is expansive, such as in these complex math, logic, and embodied planning tasks, DFS may sink into inappropriate reasoning subspace with the first steps, thereby failing to explore the full space. In contrast, for tasks with a limited search space, such as Game-24, DFS doesn't hamper the exploration.

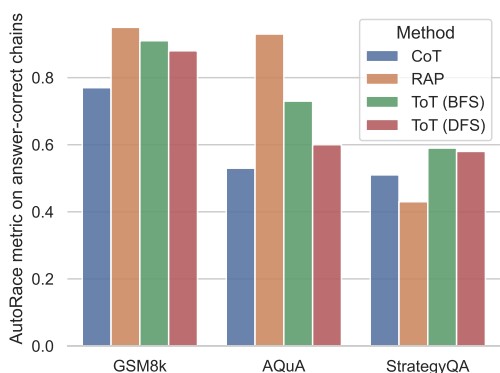

**Crucial Role of World Model in Embodied Tasks.** RAP stands out as the most effective method across most datasets, thanks to its explicit world model. This enables the LLM to predict and track state changes during reasoning, allowing for decisions based on the current state. Specifically, it outperforms ToT by 42% on Blocksworld. As previous research suggests (Xiang et al., 2023), LLMs miss essential embodied

Figure 4: `AutoRace` metric of different reasoning methods on answer-correct chains. We find search-based methods, ToT and RAP have higher `AutoRace` scores, indicating fewer false positive reasoning chains.

knowledge and skills, e.g., tracking objects over long action sequences. Thus, an explicit world model that maintains the current state would greatly reduce the difficulties in memorizing or reasoning about previous actions in an embodied environment.

**Impact of Prompt Format on False Positives.** Interestingly, StrategyQA witnesses a higher false positive rate of RAP (Figure 4). Based on errors identified by `AutoRace`, we discovered

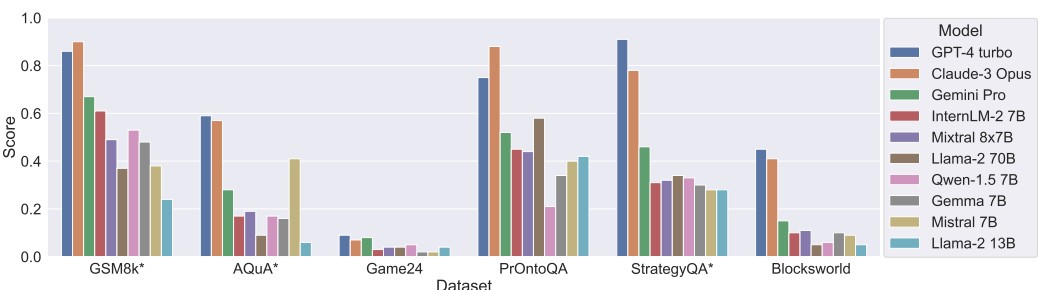

Figure 5: Results of various LLMs using CoT on every dataset. We apply `AutoRace` on three datasets with *, and oracle verifiers on other datasets. On three datasets marked with ∗, LLMs are ordered by average performance.

a common failure mode from the reasoning chains generated by RAP: The prompt design that guides LLMs to iteratively ask and answer sub-questions encourages LLMs to generate excessive details. This makes it easier to introduce factual errors but does not necessarily affect the accuracy of final answers. For example, for the problem presented in Appendix C.2, RAP raises the incorrect number 7,000 in the explanation, which is identified as an error by `AutoRace`. Conversely, CoT avoids this pitfall by not delving into unnecessary details. It's worth noting that, this prompt format is not a problem for math reasoning tasks, including GSM8k and AQuA, because every detail needs to be accurate to solve a math problem. This suggests that prompt design should be tailored to the task domain.

## 5.3 Evaluating Leading LLMs

We also use the same experimental setting to compare the step-by-step reasoning ability of multiple popular LLMs, including GPT-4 (OpenAI, 2023), Claude-3 Opus[2], Gemeni pro (Team et al., 2023), InternLM-2 (Cai et al., 2024), Mistral (Jiang et al., 2023), Mixtral (Jiang et al., 2024), Llama-2 (Touvron et al., 2023b), Qwen (Bai et al., 2023), and Gemma (Team et al., 2024). The overall results are shown in Figure 5, with more details in Table 11

**Overall Rankings.** GPT-4 turbo and Claude-3 Opus are the two with the strongest reasoning abilities, and they lead on almost every reasoning task. Surprisingly, InternLM-2 7B surpasses much larger models (e.g., Llama-2 70B) on average performance. We also notice the ranking of Top-3 models is aligned with ChatArena leaderboard[3], which indicates that the reasoning ability is indeed crucial to power the SOTA chatbot.

**Reasoning Tasks.** Top models have achieved remarkable performance on math word problems (GSM8k) and commonsense reasoning (StrategyQA), but reasoning tasks that require strong planning abilities (e.g., Game-24 and Blocksworld) remain unsolved, which leaves room for future research. Interestingly, on StrategyQA, the answer accuracy of different models is similar (0.63 - 0.79), but the `AutoRace` results differ a lot, ranging from 0.28 to 0.91, with a totally different ranking. Further examination reveals the questions in StrategyQA are often ambiguous and overly simplified. GPT-4, Claude-3, and Gemini demonstrate a more thorough consideration of these problems, unlike other models (and sometimes even the ground truth reasoning chains) which can suffer from baseless assumptions and flawed logic. The difference might be attributed to the RLHF process, which aligns model response to human preference.

## 6 Conclusion

We propose `AutoRace`, LLM-powered automated evaluation of reasoning chains, and `LLM Reasoners`, a unified formulation and library for diverse step-by-step reasoning algorithms. On this basis, we conducted comprehensive experiments to analyze the factors contributing to reasoning. In the future, it will be interesting to extend `LLM Reasoners` by including more algorithms, supporting fine-tuning methods natively, exploring multi-modal reasoning, and we plan to apply `AutoRace` for broader comparison.

---

[2]`https://www.anthropic.com/news/claude-3-family`
[3]`https://huggingface.co/spaces/lmsys/chatbot-arena-leaderboard`

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

| Method | Math | | Common | | Logical | |
|---|---|---|---|---|---|---|
| Dataset | GSM8k | Arith | Strategy | Cosmos | Logic | Sort |
| $Correct_{ans}$ | 110 | 45 | 100 | 82 | 45 | 45 |
| $Incorrect_{ans}$ | 90 | 255 | 100 | 97 | 255 | 255 |
| $Correct_{human}$ | 101 | 62 | 71 | 31 | 6 | 34 |
| $Incorrect_{human}$ | 99 | 238 | 129 | 148 | 294 | 266 |
| FP rate | 0.16 | 0 | 0.39 | 0.67 | 0.87 | 0.33 |

Table 4: The distribution of datasets for evaluation metrics.

| Method | Math | | Common | | Logical | |
|---|---|---|---|---|---|---|
| Dataset | GSM8k | Arith | Strategy | Cosmos | Logic | Sort |
| Answer-based | 0.89 | 0.99 | 0.76 | 0.63 | 0.87 | 0.94 |
| SocREval | 0.89 | 0.92 | 0.96 | 0.91 | 0.89 | 0.76 |
| Direct (trace) | 0.90 | 0.34 | 0.78 | 0.74 | 0.21 | 0.33 |
| Direct (step) | 0.96 | 0.44 | 0.85 | 0.87 | 0.75 | 0.32 |
| CoT (step) | 0.93 | 0.88 | 0.78 | 0.84 | 1.00 | 0.87 |
| AutoRace (Ours) | 0.91 | 0.90 | 0.82 | 0.84 | 0.97 | 0.87 |

Table 5: Evaluation accuracy for reasoning chains labeled incorrect by humans.

# A  Additional Details of AutoRace

## A.1  Dataset Statistics

For all datasets we experiment on, we list the statistics by answer labels and human labels in Table 4. We also include the false positive (FP) rate, defined as the proportion of instances with correct answers but labeled as incorrect by human evaluators, relative to the total number of instances with correct answers.

## A.2  Results on Reasoning Chains of Different Human Labels

We calculate the accuracy on two subsets: (1) reasoning chains labeled correct by human, and (2) reasoning chains labeled incorrect by human. The results are listed in Table 5 and Table 6. Generally, AutoRace balances the performance on both datasets, while SocREval suffers from a severe drop on the reasoning chains labeled incorrect.

| Method | Math | | Common | | Logical | |
|---|---|---|---|---|---|---|
| Dataset | GSM8k | Arith | Strategy | Cosmos | Logic | Sort |
| Answer-based | 0.99 | 0.71 | 0.76 | 0.87 | 1.00 | 0.88 |
| SocREval | 0.89 | 0.55 | 0.24 | 0.29 | 1.00 | 0.85 |
| Direct(trace) | 0.89 | 0.58 | 0.83 | 0.58 | 1.00 | 0.68 |
| Direct(step) | 0.74 | 0.40 | 0.78 | 0.03 | 0.83 | 0.38 |
| CoT(step) | 0.82 | 0.11 | 0.78 | 0.13 | 1.00 | 0.14 |
| AutoRace (Ours) | 0.90 | 0.85 | 0.72 | 0.45 | 1.00 | 0.74 |

Table 6: Evaluation accuracy for reasoning chains labeled correct by humans.

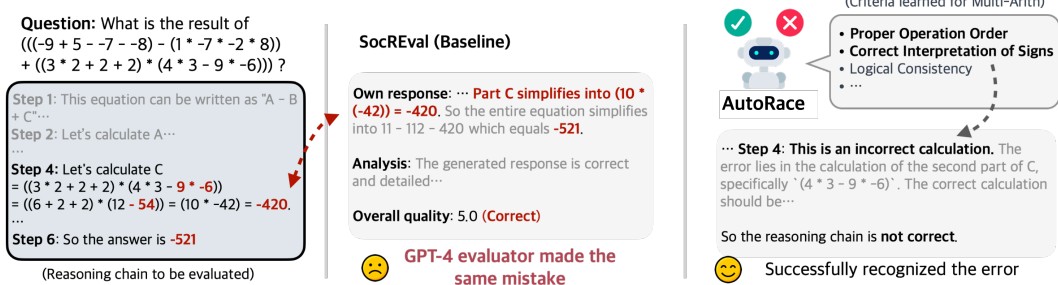

Figure 6: A case study on MultiArith (Srivastava et al., 2023). SocREval (He et al., 2023) requires GPT-4 to generate its own response to the problem, and then use it as the reference to evaluate the reasoning chain. In this case, it makes the same mistake as the reasoning chain. AutoRace successfully recognized the calculation error, guided by the criterion list learned for this task.

| Method | Word-Sort (Long) | StrategyQA (Short) |
|---|---|---|
| SocREval | 1686 / 427 / $0.030 | 357 / 269 / $0.012 |
| Direct (trace) | 2265 / 1 / $0.023 | 507 / 1 / $0.005 |
| Direct (step) | 17517 / 12 / $0.176 | 1349 / 4 / $0.014 |
| CoT (step) | 26839 / 5504 / $0.434 | 2688 / 575 / $0.044 |
| AutoRace (Ours) | 1382 / 435 / $0.027 | 270 / 389 / $0.014 |

Table 7: Average cost on one problem for different evaluation methods, in the form of (input token number / output token number / cost per question).

### A.3  Case Study

In Figure 6, we show a case study of AutoRace on MultiArith (Srivastava et al., 2023), in comparison with SocREval (He et al., 2023).

### A.4  Comparison on the Cost of Reasoning Chain Evaluation Metrics

Table 7 shows the average input token number, output token number, and the cost per question using different evaluation methods. We list the statistics on two tasks: Word-Sort, which usually has long reasoning chains, and StrategyQA, which has shorter reasoning chains.

### A.5  Example Number of Criteria List Construction

For each task in Table 11, we use 4 incorrect reasoning chains to create a criterion list. Here, we further explored different sample sizes (2, 4, 10) in Table 8.

The results indicate that AutoRace is relatively robust to the sample set size. We found that it's because:

| Method | Math | | Common | | Logical | | Average |
|---|---|---|---|---|---|---|---|
| | GSM8k | Arith | Strategy | Cosmos | Logic | Sort | |
| AutoRace (2) | 0.88 | 0.84 | 0.84 | 0.75 | 0.96 | 0.86 | 0.85 |
| AutoRace (4) | 0.91 | 0.85 | 0.79 | 0.78 | 0.97 | 0.86 | 0.86 |
| AutoRace (10) | 0.86 | 0.82 | 0.84 | 0.77 | 0.98 | 0.89 | 0.86 |

Table 8: The performance of AutoRace with different number of incorrect reasoning chains (2, 4, 10) to construct the criteria list.

| Method | Math | | Common | | Logical | | Average |
|---|---|---|---|---|---|---|---|
| | GSM8k | Arith | Strategy | Cosmos | Logic | Sort | |
| AutoRace | 0.91 | 0.85 | 0.79 | 0.78 | 0.97 | 0.86 | 0.86 |
| SocREval | 0.89 | 0.85 | 0.71 | 0.80 | 0.89 | 0.77 | 0.82 |
| SocREval (GSM8k) | 0.89 | 0.84 | 0.63 | 0.74 | 0.85 | 0.81 | 0.79 |
| AutoRace (GSM8k) | 0.91 | 0.83 | 0.73 | 0.69 | 0.99 | 0.89 | 0.84 |

Table 9: The performance of AutoRace and SocREval (He et al., 2023) using the criteria list/prompt for GSM8k.

- Besides concrete errors in examples, GPT-4 can also infer some evaluation criteria from the problem domain. E.g., For math word problems, even if there is no calculation error in the examples, GPT-4 may supplement a criterion about "accurate calculation".

- GPT-4 has certain prior knowledge on how to evaluate a reasoning chain. Thus, even if a criterion list misses some items, GPT-4 still has a chance to correctly evaluate a reasoning chain with that type of error.

### A.6 Generalization of Criteria List

To understand how general the criteria list constructed by AutoRace is, we test with the AutoRace criterion list of GSM8k on all tasks. We also test SocREval (He et al., 2023) on all tasks with its prompt for GSM8k. As shown in Table 9, AutoRace shows good generalization across different reasoning tasks.

### A.7 AutoRace Prompt

In this section, we demonstrate the pipeline of AutoRace for GSM8k (Cobbe et al., 2021) as examples. For other reasoning tasks, the template is the same, and only the questions, student answers, and extracted criteria need to be replaced.

**Prompt for detection and summarization**

```
You are a teacher.  Below are some questions, reference answers and the
answers from students.

Question:
Janet's ducks lay 16 eggs per day.  She eats three for breakfast every
morning and bakes muffins for her friends every day with four. She sells the
remainder at the farmers' market daily for $2 per fresh duck egg. How much
in dollars does she make every day at the farmers' market?
Reference answer:
Janet sells 16 - 3 - 4 = 9 duck eggs a day. She makes 9 * 2 = 18 every day
at the farmer's market.
Student answer:
Janet's ducks lay 16 eggs per day. She eats three for breakfast every morning
and bakes muffins for her friends every day with four. This means she uses
3 + 4 = 7 eggs every day. She sells the remainder at the farmers' market
daily for $2 per fresh duck egg. So she sells (16 - 7) * $2 = $6 worth of
eggs every day. The answer is 6.

Question:
Claire makes a 3 egg omelet every morning for breakfast. How many dozens of
eggs will she eat in 4 weeks?
Reference answer:
```

She eats 3 eggs every day and there are 7 days in a week so she eats 3*7 = 21 eggs a week After 4 weeks she will have eaten 4*21 = 84 eggs There are 12 eggs in 1 dozen and she'll eat 84 eggs so that's 84/12 = 7 dozen eggs.
Student answer:
Claire makes a 3 egg omelet every morning. In one week she will eat 3 * 7 = 21 eggs. In 4 weeks she will eat 4 * 21 = 84 eggs. The answer is 84.

Question:
Gloria is shoe shopping when she comes across a pair of boots that fit her shoe budget. However, she has to choose between the boots and two pairs of high heels that together cost five dollars less than the boots. If one pair of heels costs $33 and the other costs twice as much, how many dollars are the boots?
Reference answer:
The second pair of heels costs 33 * 2 = $66. The heels together cost 66 + 33 = $99. The boots cost $5 more than both pairs of heels together, so the boots cost 99 + 5 = $104.
Student answer:
We know that one pair of heels costs $33 and the other pair costs twice as much. This means that the other pair costs $33 * 2 = $66. Together, the two pairs of heels cost $33 + $66 = $99. The boots cost five dollars less than the heels, so the boots cost $99 - $5 = $94. The answer is $94.

Question:
Mark's car breaks down and he needs to get a new radiator. The cost for a new radiator is $400 but he goes to get it at a junk shop and gets it for 80% off. He then hires a mechanic to install it and it takes 3 hours at $50 an hour. How much did he pay?
Reference answer:
The discount on the radiator was 400*.8=$320 So he paid 400-320=$80 The mechanic charges 3*50=$150 So in total he paid 80+150=$230
Student answer:
The cost for a new radiator is $400. He got it at a junk shop for 80% off, so he paid $400 * 0.8 = $320 for the radiator. The mechanic charged him $50 per hour for 3 hours, so he paid $50 * 3 = $150 for the labor. In total, he paid $320 + $150 = $470. The answer is 470.

Please summarize the mistakes in a short sentence for the question. At the end, please make a brief list of criteria. Make sure they are general and not specific to these questions so that others can grade the answers for other answers by following these criteria.

**Prompt for Evaluation**

Below is a question and an answer from a student. You are required to check the correctness of the reasoning chains step by step. The criteria are as follows:

**Accuracy in Mathematical Operations:** Ensure calculations are correct and follow logical mathematical principles.
**Understanding the Problem Statement:** Comprehend the details and conditions of the question accurately.
**Correct Application of Mathematical Concepts:** Apply the right mathematical formulas, operations, or concepts to solve the problem.
**Unit Conversion and Appropriateness:** When required, correctly convert units and use appropriate units in the answer.
**Final Answer Relevance:** Ensure the final answer directly addresses the question asked, and is presented clearly and concisely.

| Type | Question | Reasoning Chain | Explanation |
|---|---|---|---|
| A | Henry made two stops during his 60-mile bike trip. He first stopped after 20 miles. His second stop was 15 miles before the end of the trip. How many miles did he travel between his first and second stops? | Step 1 - Henry traveled 20 miles + 15 miles = 35 miles **between his first and second stops**. Step 2 - Henry traveled 60 miles - 35 miles = 25 miles without stopping. Step 3 - The answer is 25 | There is a hallucination in an early reasoning step, but the following reasoning steps ignore the mistake and reach the correct answer. |
| B | Is "A Tale of Two Cities" a parody of the Bible? | "A Tale of Two Cities" was published in 1859, and the Bible was published much earlier. **Thus**, "A Tale of Two Cities" is not a parody of the Bible. So the answer is no. | The reasoning chain has obvious or multiple mistakes, but hits the correct answer by chance. |
| C | Is clementine pith highly sought after? | **Clementine pith is not highly sought after.** So the answer is no. | The reasoning chain is not informative at all, though the answer is correct. |

Table 10: Common types of false positive reasoning chains detected by AutoRace. The example questions are from StrategyQA (Geva et al., 2021) and GSM8k (Cobbe et al., 2021), and the reasoning chains are generated by Llama-2 70B. An explanation for each type of false positive reasoning chain is attached.

```
**Logical Reasoning and Step-by-Step Explanation:** The answer should
include a logical, step-by-step explanation that demonstrates how the final
answer was reached.

Question:
[QUESTION]

Student answer:
[INPUT]

Please check the answer through each criterion, and make sure you carefully
examine each reasoning step. Finally, if there is any step that fails the
verification, output an INCORRECT, or else output a CORRECT.
```

The criteria are provided by the GPT in the Summarization step.

## A.8 False Positive Reasoning Chains Detected by AutoRace

AutoRace can serve as a scalable way to detect false positive reasoning chains. Here, we summarize several types of false positive reasoning chains in Table 10.

## A.9 Pseudo-code of AutoRace

We present the pseudo-code of AutoRace in Algorithm 1.

---

**Algorithm 1** Criteria list construction of `AutoRace`

---

1: **Input:** Student LLM $f_{Student-LLM}$, GPT-4 $f_{GPT-4}$, a training set of questions, reference answers, and reasoning chains $D_{train} = \{x, y_r, z_r\}$, a prompt template to create criterion $T_{criterion}$, the expected number of error examples $n$
2: **Output:** Criterion list $C$
3: $D \leftarrow \text{Sample}(D_{train})$         ▷ Initialize the incorrect reasoning chain set
4: $D_{error} \leftarrow \text{EmptySet}$       ▷ Randomly sample a subset of the training set
5: **for** $(x, y_r, z_r)$ **in** $D$ **do**
6:   $y, z = f_{Student-LLM}(x)$        ▷ Answer the question with the student LLM
7:   **if** $y \neq y_r$ **then**
8:    Add $\{x, y_r, z_r, z\}$ to $D_{error}$       ▷ Collect incorrect reasoning chains
9:    **if** $|D_{error}| \geq n$ **then**
10:     **break**          ▷ Already enough error examples
11:    **end if**
12:   **end if**
13: **end for**
14: $C = f_{GPT-4}(D_{error}, T_{criterion})$      ▷ Let GPT-4 summarize from error examples

---

```python
from reasoners import SearchConfig, WorldModel, Reasoner
from reasoners.algorithm import MCTS
from reasoners.lm import Llama2Model

class MyWorldModel(WorldModel):
    def step(self, state, action):
        return self.llm.generate(self.next_state_prompt.format(state, action))
    ...

class MyConfig(SearchConfig):
    def reward(self, state, action):
        return self.llm.generate(self.eval_prompt.format(state, action))
    ...

reasoner = Reasoner(world_model=MyWorldModel(), search_config=MyConfig(), search_algo=MCTS())
```

Figure 7: The three key components in a reasoning algorithm, *reward function*, *world model*, and *search algorithm* in the formulation (top), correspond to three classes in `LLM Reasoners`. To implement a reasoning algorithm for a certain domain (a Reasoner object), a user may inherit the `SearchConfig` and `WorldModel` class, and import any `SearchAlgorithm`.

## B   Details of `LLM Reasoners`

**World Model.** The `WorldModel` class is responsible for managing all the state changes during reasoning. It includes `init_state` to create the initial state, `step` to predict the next states, and `is_terminal` to identify terminal states. Utilizing our consistent API, users can effortlessly implement a world model for a specific task, or adapt the default world model recording previous actions, as done in CoT (Wei et al., 2022), ToT (Yao et al., 2023), etc.

**Search Configuration.** The `SearchConfig` class mainly includes two important functions: `get_actions` to decide the action space under each state to facilitate searching, and `reward` to assess the quality of each reasoning step.

**Search Algorithm.** The `SearchAlgorithm` specifies the strategy to explore the reasoning space. We have implemented several search algorithms in our library, e.g., Greedy Decoding (Khalifa et al., 2023), Beam Search (Xie et al., 2023; Creswell & Shanahan, 2022), Depth-first Search (Yao et al., 2023), and Monte-Carlo Tree Search (Hao et al., 2023a). These algorithms are designed to work seamlessly with any world model and search configuration on any reasoning task.

**Other Features.** `LLM Reasoners` has integrated To power these modules above with LLMs conveniently, we offer a standardized interface `LanguageModel` that supports a range of LLM libraries, including HuggingFace transformers (Wolf et al., 2020), `facebookresearch/llama` (Touvron et al., 2023a;b), APIs of GPT (OpenAI, 2023), Claude and Gemini (Team et al., 2023). We also integrated libraries specialized in quantization, like Exllama (Frantar et al., 2022), to reduce the hardware requirements. Additionally, the `Benchmark` class provides a

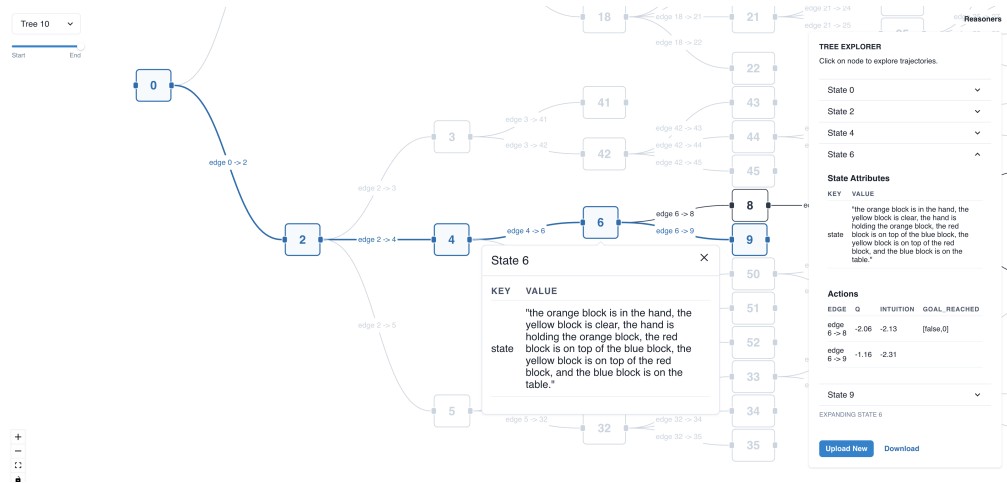

Figure 8: A screenshot of the visualization tool to diagnose the RAP algorithms on the Blocksworld task.

standard platform (e.g., standard prompts, evaluation methods) for a collection of widely recognized reasoning tasks, such as GSM8k (Cobbe et al., 2021), StrategyQA (Geva et al., 2021), and Blocksworld (Valmeekam et al., 2023). We also include `Visualizer`, an interactive visualization tool that allows for the straightforward depiction of the search trees. This tool significantly lowers the complexity of developing and analyzing the complicated reasoning process. We show a screenshot of the visualization module of `LLM Reasoners` in Figure 8. Users have the option to upload the log file to our server with one line of code and use the web-based visualization to debug the reasoning algorithms. It allows users to interact with the log by selecting different reasoning trees, and checking the details of each node and edge in the tree. This tool significantly lowers the difficulty of applying a complex reasoning algorithm to new tasks.

## C  Additional Details of Experiment Analysis

### C.1  Details of RAP Implementation

We explicitly define states and world models in RAP. For free-form question-answering tasks, including GSM8k, AQuA, and StrategyQA, a state is considered as the set of known variables. An action involves posing a sub-question about an unknown variable. The world model then responds to this sub-question, thereby augmenting the state with the value of a new intermediate variable. For tasks with a clearly defined intermediate state, such as the remaining numbers in Game of 24, the latest facts in PrOntoQA, and the current block arrangement in Blocksworld, we directly employ these intermediate states in natural language description. Actions in these scenarios are defined respectively as a basic arithmetic operation, a one-hop logic reasoning, and a single block movement. The world model is prompted to predict the subsequent state as per the task's definition.

In GSM8k, AQuA, and StrategyQA, we use the rewards from the confidence of state transition and self-evaluation of usefulness of each step. In Game of 24, we use the rewards from self-evaluation of the possibility to achieve 24 for the state after each step. In PrOntoQA, we use the rewards of action likelihood and self-evaluation of the correctness of each step. In Blocksworld, we use the rewards from the action likelihood and task-specific heuristics, i.e., the number of achieved subgoals.

### C.2  Case Study on Prompt formats

Below, we present the reasoning chain of RAP and CoT to a question in StrategyQA.

| Model | GSM8k | AQuA | Game24 | Pronto | Strategy | Block | Average |
|---|---|---|---|---|---|---|---|
| GPT-4 turbo | 0.86 (0.93) | 0.59 (0.71) | 0.09 | 0.75 | 0.91 (0.65) | 0.45 | 0.66 |
| Claude-3 Opus | 0.90 (0.90) | 0.57 (0.79) | 0.07 | 0.88 | 0.78 (0.63) | 0.41 | 0.66 |
| Gemini pro | 0.67 (0.72) | 0.28 (0.48) | 0.08 | 0.52 | 0.46 (0.71) | 0.15 | 0.45 |
| InternLM-2 7B | 0.61 (0.68) | 0.17 (0.45) | 0.03 | 0.45 | 0.31 (0.69) | 0.10 | 0.39 |
| Llama-2 70B | 0.37 (0.54) | 0.09 (0.34) | 0.04 | 0.58 | 0.34 (0.76) | 0.05 | 0.35 |
| Qwen-1.5 7B | 0.53 (0.56) | 0.17 (0.28) | 0.05 | 0.21 | 0.33 (0.79) | 0.06 | 0.33 |
| Gemma-7B | 0.48 (0.57) | 0.16 (0.34) | 0.02 | 0.34 | 0.30 (0.66) | 0.10 | 0.33 |
| Mistral 7B | 0.38 (0.41) | 0.12 (0.32) | 0.02 | 0.40 | 0.28 (0.66) | 0.09 | 0.33 |
| Llama-2 13B | 0.24 (0.29) | 0.06 (0.20) | 0.04 | 0.42 | 0.28 (0.66) | 0.05 | 0.25 |

Table 11: The comparison of different models across various tasks. GSM8k, AQuA and Strategy are evaluated with `AutoRace` (The answer accuracy is shown as a reference in the bracket). Other results use the oracle verifier.

```
——Question——
Did either Kublai Khan or his grandfather practice monogamy?
——RAP———
Step 1: Did Kublai Khan have a harem? Kublai Khan had a harem of 7,000 women.
So the answer is yes.
Step 2: Did Genghis Khan have a harem? Genghis Khan had a harem of 500 women.
So the answer is yes.
Step 3: Does having a harem of women mean practicing polygamy? Having a
harem of women means practicing polygamy. So the answer is yes.
Step 4:  Now  we  can  answer  the  question:  Did  either  Kublai  Khan  or
hisgrandfather practice monogamy? Kublai Khan had a harem of 7,000 women.
Genghis Khan  had  a  harem  of  500  women.   Having  a  harem  of  women  means
practicing polygamy. Thus, neither Kublai Khan nor his grandfather practiced
monogamy. So the answer is no.
——CoT———
Step 1: Both Kublai Khan and his grandfather, Genghis Khan, were polygamous.
Step 2: Thus, neither Kublai Khan nor his grandfather practiced monogamy.
Step 3: So the answer is no.
——reference—
Step 1: Kublai Khan was married multiple times and was said by some to have
thousands of concubines.
Step 2: Kublai Khan's grandfather was Genghis Khan.
Step 3: Genghis Khan had six wives and was said to have over 500 concubines.
Step 4: So the answer is no.
```

## C.3 `AutoRace` Leaderboard

We show the detailed evaluation results on leading LLMs in Table 11.

