# OpenReview forum: "LLM Reasoners: New Evaluation, Library, and Analysis of Step-by-Step Reasoning with Large Language Models"
_colmweb.org/COLM/2024/Conference — COLM_

### Official Review · Reviewer_bv3g · 2024-05-08

**Rating:** 6
**Confidence:** 3
**Ethics Flag:** 1

**Summary:**

This paper presents a new evaluation package of LLMs' complex reasoning capabilities. This paper has several contributions: (1) A new automatic evaluation metric, which first asks GPT4 to conclude common failure reasons of wrong responses on the trainined set and then ask GPT4 to evaulate models' output based on these reasons to achieve automatic evaluation. (2) A configurabale toolkit that allows user to specify their foundation model and reasoning strategy. In the end, this paper also presents a comprehensive evaluation of the peformance of llama2-70B with several reasoning strategies (i.e., CoT, RAP, and ToT)

**Questions To Authors:**

1. In the fourth paragraph of introduction, the paper mentrions that AutoRace and detect 70% of the incorrect reasoning chains. Can you elaborate more on that? This seems to be a moderate agreement.
2. Can you give a more clear definition of the mentioned world model? like what kind of world model can be supported by the current syste. I found it missing the first time this paper discussing the world model.
3. Regarding the criteria list, is it generalizable to other tasks?
4. Regarding the error collection, how to guarantee the coverage of the collected errors? Better to add an experiment to support this since this is the foundation of AutoRace.
5. In table 5, AutoRace is annotated as fully auto. However, if my understanding is correct, it still requires the support of a set of examples to construct the criteria list. Can you elaborate more on that?

**Reasons To Accept:**

1. This paper proposes a new evaluation system for LLMs' reasoning capabilities.
2. The comprehensive evaluation reveals certain limitations of existing LLMs.

**Reasons To Reject:**

The proposed method is hard to generalize. Based on the description in the paper, the proposed solution's generalization capability remains unclear. For example, the current system requires a set of questions (i.e., training data) to sample failure response for each task and then conclude the criteria list. As a result, this list might be only applicable to the current dataset. It would be great if some experiments can support the genralization capabilities of the criteria list.
Regarding the unified library, it is unclear how to support reasoning search algorithm that requires the feedback from RMs since these models often require additional task-specific training.

---

> ### Author Rebuttal · Authors · 2024-05-31
>
> Thanks for acknowledging that our new evaluation reveals certain limitations of existing LLMs!
>
> **Task-specific training**
>
> There are various open-source libraries to train reward models (RMs), e.g., Transformers, TRL, etc. Our library focuses on inference-time algorithms and is orthogonal to them. LLM Reasoners allows for flexible implementation of the reward function, including using a trained RM, and users can easily call any algorithms to search for high-reward reasoning chains. In the future, we plan to support RM training natively with our library.
>
> **70% incorrect reasoning chains**
>
> AutoRace recognizes 70% of the **false positive** reasoning chains. These are more challenging to identify than usual incorrect chains, because their errors are often subtle, and don’t affect the final answer. With traditional answer-based evaluation, **NONE** of these chains would be detected. Discovering 70% of them already led to multiple novel findings (Section 5.2)
>
> To further clarify, AutoRace can detect **~90%** of all incorrect reasoning chains across 6 datasets (Figure 3), and we have shown it has higher accuracy and better robustness compared to various baselines with a comprehensive evaluation in Section 3.
>
> **World model**
>
> A world model $P_\theta$ predicts state transition: $s_t \sim P_\theta (s_t | s_{t−1}, a_t)$. LLM Reasoners allows users to flexibly implement the world model, e.g., For RAP on the embodied task Blocksworld, the world model is implemented as a function that calls an LLM to predict the new arrangement of blocks after an action.
>
> **Generalization**
>
> We test AutoRace with the criterion lists of GSM8k on all tasks. We also test SocREval on all tasks with its prompt for GSM8k. AutoRace shows a high degree of generalization. We found that it’s because many of the criteria discovered by AutoRace are general among different reasoning tasks.
>
> |Method|GSM8k|Arith|StrategyQA|Cosmos|Logic|Sort|Average|
> |-|-|-|-|-|-|-|-|
> |AutoRace|0.91|0.85|0.79|0.78|0.97|0.86|0.86|
> |SocREval|0.89|0.85|0.71|0.80|0.89|0.77|0.82|
> |SocREval (GSM8k)|0.89|0.84|0.63|0.74|0.85|0.81|0.79|
> |AutoRace (GSM8k)|0.91|0.83|0.73|0.69|0.99|0.89|0.84|
>
> **Error collection and example set**
>
> Due to the word limit, please refer to our response to reviewer SuhS for additional experiment results.
>
> Thanks again for your thoughtful review! We will update our paper to reflect the additional results and discussion.

---

> > ### Comment · Reviewer_bv3g · 2024-06-02
> >
> > Thanks for the reply.
> >
> > I have increased the overall rating from 5 to 6.
> >
> > Your results on GSM8K are encouraging. However, I still think designing critics based on a dataset is not the most convincing way because we cannot know what type the user query belongs to in a real system. I know it is hard, but trying to think out of specific tasks and find more fundamental critics (or letting the model identify by itself) could make the system more helpful in real applications.

---

> > > ### Author Response · Authors · 2024-06-04
> > > **Re: Official Comment by Reviewer bv3g**
> > >
> > > Thank you for your feedback!
> > >
> > > Our current work focuses on analyzing LLM reasoning, and constructing an automatic reasoning chain evaluation metric, even though tailored to several reasoning datasets, takes a concrete step towards this objective (e.g., it helps reveal the novel conclusions in Section 5).
> > >
> > > We recognize the importance of developing an evaluation or reward model for general purposes. Such a model remains a goal for our long-term research efforts, and we are dedicated to advancing towards it. For example, a natural next step of AutoRace would be using human feedback to learn a better criteria list that applies to general real-world applications.

---

### Official Review · Reviewer_SuhS · 2024-05-11

**Rating:** 7
**Confidence:** 4
**Ethics Flag:** 1

**Summary:**

This paper presents principled methods on how to organize automatic evaluation of reasoning chains (AutoRace) and design a tailored evaluation library with a unified view of LLM reasoning algorithms. The authors conduct an extensive study on various configurations, including the choice of LLMs and promptings. Their experiments reveal some interesting findings, e.g., search breadth is superior to depth in LLM reasoning.

**Reasons To Accept:**

1. The proposed AutoRace is fully automatic and can achieve comparable performance to baselines with expert-crafted annotations.
2. The unified formulation of LLM reasoning is a good principle for implementing evaluation systems and allows for a systematic comparison across reasoning algorithms.
3. Extensive experiments suggest intriguing observations and provide best practices and insight for LLM reasoning. These experiments also validate the efficacy of their system.

**Reasons To Reject:**

1. Novelty could be somewhat limited, but the system development is intensive and the software could be useful for the community.
2. AutoRace relies heavily on the availability of training sets. However, constructing such a training set for a specific reasoning task is expensive and contradicts the main purpose of AutoRace, which aims to reduce human efforts. This weakens the claim of this paper.

---

> ### Author Rebuttal · Authors · 2024-05-31
>
> Thank you for recognizing that AutoRace is fully automatic and performant, LLM reasoning is a good principle, and the experiments suggest intriguing observations about LLM reasoning!
>
> **Novelty**
>
> We’d like to highlight 3 novel contributions of this work:
> - We propose a novel reasoning chain evaluation method and show the promise to be an important metric complementing answer-based evaluation.
> - We present a novel perspective to unify numerous current reasoning methods, and build a library for developing, deploying, and analyzing reasoning algorithms.
> - Our comprehensive experiments lead to several novel findings about LLM reasoning:
>   - Reward-guided search reduces false positives
>   - Search breadth matters more than depth
>   - World model is crucial for embodied tasks
>   - Inappropriate prompt format may lead to false positives
>
> **Reliance on training set**
>
> While we understand this is a reasonable concern, we argue that most reasoning datasets already include a training set, which eliminates the need for specific construction for AutoRace. Furthermore, as AutoRace can be effective with a very **small** demonstration set, which we will show below, even if the training set is not available, it’s easy to find or create a few demonstrations and apply AutoRace.
>
> For each task in Table 1, we use only 4 wrong reasoning chains to create a criterion list. Here, we further explored different sample sizes (2, 4, 10).
>
> |Method|GSM8k|Arith|StrategyQA|Cosmos|Logic|Sort|**Average**|
> |-|-|-|-|-|-|-|-|
> |AutoRace (2)|0.88|0.84|0.84|0.75|0.96|0.86|**0.85**|
> |AutoRace (4)|0.91|0.85|0.79|0.78|0.97|0.86|**0.86**|
> |AutoRace (10)|0.86|0.82|0.84|0.77|0.98|0.89|**0.86**|
>
> The results indicate that AutoRace is relatively robust to the sample set size. We found that it’s because:
>
> - Besides concrete errors in examples, GPT-4 can also infer some evaluation criteria from the problem domain. E.g., For math word problems, even if there is no calculation error in the examples, GPT-4 may supplement a criterion about “accurate calculation”.
> - GPT-4 has certain prior knowledge on how to evaluate a reasoning chain. Thus, even if a criterion list misses some items, GPT-4 still has a chance to correctly evaluate a reasoning chain with that type of error.
>
> Additionally, our tests confirm that the criteria list generalizes well across different reasoning tasks (See new results in our response to reviewer bv3g). This further reduces the dependency on task-specific demonstration data.

---

> > ### Comment · Reviewer_SuhS · 2024-06-02
> > **Re: Rebuttal by Authors**
> >
> > Thanks for your response! The additional experiment addresses my concern about the sample efficiency of AutoRace. However, I noticed an interesting phenomenon in the new result, where providing more incorrect reasoning chains to GPT-4 does not yield a consistent gain (at least for 2/6 datasets). Do you have any explanation for that?

---

> > > ### Author Response · Authors · 2024-06-04
> > > **Re: Re: Rebuttal by Authors**
> > >
> > > Thanks for your reply!
> > >
> > > One possible reason is that the criteria list constructed by GPT-4 given more examples includes some general criteria, such as `relevant to the question` . These criteria may not be specific enough to help improve evaluation. How to make use of more examples to consistently improve evaluation accuracy remains an interesting problem, and we plan to explore it in future work. For example, it's possible to use a small validation set to select the best composition of criteria, or even find the best criteria list with some tree search algorithms [1].
> > >
> > > [1] Wang et al., 2023, "PromptAgent: Strategic Planning with Language Models Enables Expert-level Prompt Optimization"

---

> > > > ### Comment · Reviewer_SuhS · 2024-06-04
> > > > **Re: Re: Re: Rebuttal by Authors**
> > > >
> > > > Thanks for your response. I have no further questions and maintain my score.

---

### Official Review · Reviewer_rws2 · 2024-05-18

**Rating:** 7
**Confidence:** 4
**Ethics Flag:** 1

**Summary:**

The paper introduces AutoRace for automated evaluation of LLM reasoning chains and LLM Reasoners, a library standardizing reasoning algorithms for systematic comparison. These tools enable detailed analysis of reasoning methods, revealing insights into factors like reward guidance, search strategies, world models, and prompt formats.

**Questions To Authors:**

-Can you give an estimate of how accurate auto race annotations are?

**Reasons To Accept:**

- The work presents an interesting framework for evaluating reasoning tasks.
- They nicely formulated the reasoning problem under a common framework.

**Reasons To Reject:**

- The steps taken in Autorace to annotate are not clear.
- The empirical findings are mostly known. However, after the rebuttal phase, I can see enough novelties in the empirical findings.
- No comparison in Autorace annotation quality.

---

> ### Author Rebuttal · Authors · 2024-05-31
>
> Thanks for acknowledging that AutoRace is an interesting evaluation framework, and LLM Reasoners nicely formulated reasoning problems under a common framework.
>
> **Steps taken in Autorace**
>
> We have described the steps of implementing AutoRace in Section 3.1, along with a concrete running example for illustration (Figure 2). Briefly, it first creates a criterion list for a reasoning task automatically, by (1) collecting a list of wrong reasoning chains, (2) asking GPT-4 to elaborate the errors in them (3) summarizing the concrete errors into a criterion list. Finally, we apply the criterion list to evaluate new reasoning chains with GPT-4.
>
> **Empirical findings most known**
>
> We respectfully disagree with this claim. In Section 5, we have presented several novel findings:
> 1. Reasoning as reward-guided search helps not only improve final accuracy, but also effectively alleviate false-positive reasoning chains;
> 2. For efficient search in the reasoning space, the breadth of search is generally more important than the depth for most tasks;
> 3. incorporating a world model that explicitly infers reasoning state would effectively improve the LLM reasoning ability, particularly for tasks in embodied environments;
> 4. inappropriate prompt format design might inadvertently lead to false positive reasoning chains.
>
> These findings are from our comprehensive analysis of reasoning methods, which rely on our new reasoning chain evaluation method and the unified library. To the best of our knowledge, these findings haven’t been discussed in previous studies.
>
> **No comparison & Accuracy of AutoRace**
>
> In Section 3, we have compared AutoRace to four baseline methods for reasoning chain evaluation. In Table 1, we show that AutoRace can achieve 86% accuracy averaged on 6 human-annotated datasets, while the best baseline can only achieve 82%. It’s also the only method that performs within 5% of the best results across all six datasets, showing outstanding robustness. We have made a more detailed comparison with Socreval [1], the strongest baseline, by visualizing the confusion matrix (Figure 3), conducting case studies (Figure 6), and discussing the source of performance improvement (Section 3.2).
>
> We hope this response adequately addresses your concerns. We are eager to engage in discussions and welcome any additional questions you may have.
>
> [1] He et al., 2023, “Large language models with the Socratic method for reference-free reasoning evaluation”

---

> > ### Author Response · Authors · 2024-06-05
> > **Looking forward to the discussion**
> >
> > Dear reviewer,
> >
> > We have dedicated ourselves to thoroughly addressing all your concerns and questions in the rebuttal. Could you please spare a moment to review our response? We are eager for your feedback and prepared to provide any further clarifications.
> >
> > Thank you for your time and consideration.

---

> > ### Comment · Reviewer_rws2 · 2024-06-05
> >
> > Thanks for clarifying my comments. Most of my questions are clarified with your detailed response. Can you shed some light on how accurate auto-race annotations are?

---

> > > ### Author Response · Authors · 2024-06-05
> > >
> > > Thank you for your feedback!
> > >
> > > The average accuracy of AutoRace is 0.86 (from the experiment results in Table 1). This is higher than the best baseline, which achieves an accuracy of 0.82. Moreover, AutoRace is the only method that performs within 5% of the best results across all six datasets, showing outstanding robustness. For more experimental results and detailed analysis, please refer to Section 3.2 in the paper.
> > >
> > > We are happy to answer any further questions you may have.

---

> > > > ### Comment · Reviewer_rws2 · 2024-06-05
> > > >
> > > > Thanks for the prompt reply. Let me clarify the question. In Table 1, AutoRace performs better than human reasoning in some instances. That is why I was asking how accurate the basis of judgment is. Can you explain why  AutoRace can be better than a human evaluator in some instances?

---

> > > > > ### Author Response · Authors · 2024-06-06
> > > > >
> > > > > In Table 1, we didn't compare AutoRace to human evaluation. In fact, we use human evaluations of reasoning chains as the **ground truth labels** (True/False) in the experiment. Instead, we include **`answer-based evaluation`** as a reference in Table 1.
> > > > >
> > > > > ---
> > > > >
> > > > > "Why can AutoRace be better than **answer-based** evaluation in some instances?"
> > > > >
> > > > > Answer-based evaluation evaluates reasoning chains solely based on whether the final answer matches the ground truth answer, disregarding the correctness of the intermediate reasoning steps. This can lead to the **false positive** problem (as demonstrated in Figure 1, a), where a reasoning chain might conclude with a correct answer but be supported by faulty reasoning. In contrast, AutoRace is specifically designed to assess the reasoning process by applying learned evaluative criteria.

---

> > > > > > ### Author Response · Authors · 2024-06-06
> > > > > >
> > > > > > Dear reviewer,
> > > > > >
> > > > > > Thanks for your time. Could you let us know if our clarification resolved your concerns? We are happy to answer any further questions you may have.

---

### Official Review · Reviewer_5tZW · 2024-05-21

**Rating:** 6
**Confidence:** 4
**Ethics Flag:** 1

**Summary:**

This paper proposes a framework (AutoRace) to automate the evaluation of large language models (LLMs) on reasoning tasks. Specifically, the framework evaluates the reasoning chains generated by LLMs. The authors have created a library to formalize the evaluation of existing reasoning tasks across various prompting techniques, such as chain-of-thought, beam search, tree-of-thought, and reasoning-via-planning.

**Reasons To Accept:**

1. The paper focuses on automating the reasoning chain evaluation task, which can be a key to standardizing LLM evaluation.
2. The authors have provided a comprehensive comparison with contemporary techniques and demonstrated the superior performance of AutoRace.

**Reasons To Reject:**

1. The authors described their evaluation algorithm but did not provide the pseudocode.
2. In the last paragraph of section 3.1, it is unclear whether the criteria list is predefined or generated by GPT-4. If GPT-4 generates the list, how is its consistency maintained?
3. It is unclear which reasoning chains the evaluator can correctly assess and which ones it misses. An ablation study is needed.
4. Are there any biases introduced by the LLM evaluator?

---

> ### Author Rebuttal · Authors · 2024-05-31
>
> Thanks for your encouraging comments that AutoRace is performant and can be a key to standardizing LLM evaluation!
>
> **Pseudocode**
>
> We attach the pseudocode of the criterion list construction to this [link](https://anonymous.4open.science/r/rebuttal-F471/autorace-algo.pdf) due to the formatting issue of OpenReivew. After the criterion list is constructed, it's straightforward to prompt GPT-4 with it to evaluate new reasoning chains.
>
> **Consistency of Criteria list**
>
> The criteria list is generated by GPT-4. To ensure the consistency of the criterion list, we divide criteria list construction into three easier subtasks for GPT-4: (1) Collecting wrong reasoning chains (2) Detecting the errors (3) Summarizing the criteria list.
>
> Besides, our manual examination reveals that the constructed criterion lists for all tasks look reasonable (See an example in Appendix D.2). Finally, the comparative experiments in Section 3 demonstrate that AutoRace consistently outperforms established baselines, affirming the reliability of the generated criterion lists.
>
> **Case study**
>
> We have listed several successful cases of AutoRace in Table 6 and Appendix G.We conduct additional case studies and collect several failure examples ([link](https://anonymous.4open.science/r/rebuttal-F471/autorace-error.pdf)). Errors made by AutoRace can be mainly attributed to (1) Incorrect understanding of questions (2) Wrong calculation (3) Being too strict on some corner cases.
>
> **Bias of LLM evaluator**
>
> In our experiments, we observed that while LLM evaluators are effective at identifying reasoning errors, they sometimes tend to misclassify correct reasoning chains as erroneous (refer to the confusion matrix in Figure 3). Compared to the previous LLM evaluation method SocREval [1], **AutoRace exhibits less bias**. This improvement is attributed to the fundamental difference in their evaluation processes. SocREval relies on the evaluator LLM to first correctly solve the problem. Once the evaluator LLM itself fails to do so, it will misclassify correct reasoning chains as wrong. In contrast, AutoRace is not constrained by the evaluator’s problem-solving capabilities. Instead, it assesses the reasoning chain based on established criteria, simplifying the evaluation task.
>
> Thanks again for your review and suggestion. We will update our paper with the pseudo-code and case study.
>
> [1] He et al., 2023, “Large language models with the Socratic method for reference-free reasoning evaluation”

---

> > ### Author Response · Authors · 2024-06-05
> > **Looking forward to discussion**
> >
> > Dear reviewer,
> >
> > We sincerely appreciate your insightful feedback once again. In response, we have diligently addressed each of your concerns and queries. Notably, we've supplemented the pseudo-code and multiple failure cases of AutoRace. Would you be able to spare some time to review our response and confirm if it adequately addresses your questions? We are eager to engage in further discussions and provide additional clarifications on any new queries you may have.
> >
> > Thank you for your time and consideration.

---

> > ### Comment · Reviewer_5tZW · 2024-06-07
> >
> > Thank you for your detailed rebuttal. You have addressed most of my concerns. However, I would like to make two additional observations:
> >
> > 1. The comments in the pseudocode for steps 3 and 4 appear to be swapped.
> > 2. Human study evaluations are necessary to establish the reliability and soundness of the criterion lists.

---

> > > ### Author Response · Authors · 2024-06-07
> > >
> > > Thanks for your reply.
> > >
> > > **comments in the pseudo-code**
> > >
> > > We have fixed the typo and updated the pseudo-code [here](https://anonymous.4open.science/r/rebuttal-F471/autorace-algo-fix.pdf).
> > >
> > > **human evaluation of criterion lists**
> > >
> > > Our experiments demonstrate a strong alignment between AutoRace and human evaluations of reasoning chains. The overall performance of AutoRace serves as an effective indicator of the quality of criterion lists. We have also uploaded all criteria lists ([link](https://anonymous.4open.science/r/rebuttal-F471/autorace-criteria.txt)) used in our experiments. The authors have manually examined them, and concluded that all of them cover diverse dimensions of evaluation and contain no obvious mistakes.
> > >
> > > Thanks again for your feedback. We will add the updated pseudo-code and all criterion lists into the paper.

---

### Decision · Program_Chairs · 2024-07-10

**Decision:**

Accept

**Comment:**

The paper presents a framework and configurable library called AutoRace for evaluating the complex reasoning capabilities of LLMs. Evaluation in AutoRace relies on training GPT4 to identify common reasoning failures based on a given training set, and then using the trained GPT4 to evaluate model outputs. The framework allows evaluating models over various reasoning tasks and multiple prompting techniques (e.g., chain of thought, beam search, tree of thought, and reasoning via planning). The authors also demonstrate AutoRace through a comprehensive evaluation of LLaMA2-70B, which reveals practical insights and performance limitations of different prompting methods.

Devising more fine-grained and comprehensive evaluation protocols for reasoning chains is a challenging problem, and crucial for improving the reasoning abilities of LLMs. Being automatic, performant (on par with baselines with expert annotations), and easily configurable, AutoRace could be a useful tool for tackling these problems.

There are two concerns about this work. First, AutoRace reliance on training data could be expensive, require human annotations, and limit the ability of the system to generalize to reasoning settings/failures not observed in training. However, the authors showed compelling evidence during the rebuttal that their method can be effective even with a small demonstration set (rather than a large training set). The second concern is regarding biases and failures of the LLM evaluator, which could influence the overall evaluation. The authors have discussed this in the paper and during the rebuttal. While the current evaluator does introduce noise, it seems to outperform the baselines, and improving it could be an interesting direction for future work.

[At least one review was discounted during the decision process due to quality]